# Endoscopy Ultrasound-Guided Biliary Drainage Using Lumen Apposing Metal Stent in Malignant Biliary Obstruction

**DOI:** 10.3390/diagnostics13172788

**Published:** 2023-08-29

**Authors:** Alessandro Fugazza, Marta Andreozzi, Alessandro De Marco, Leonardo Da Rio, Matteo Colombo, Marco Spadaccini, Silvia Carrara, Marco Giacchetto, Mrigya Sharma, Vincenzo Craviotto, Anita Busacca, Chiara Ferrari, Alessandro Repici

**Affiliations:** 1Division of Gastroenterology and Digestive Endoscopy, Humanitas Research Hospital–IRCSS, Via Manzoni 56, Rozzano, 20089 Milan, Italy; alessandro.fugazza@humanitas.it (A.F.); marta.andreozzi@humanitas.it (M.A.); alessandro.demarco@humanitas.it (A.D.M.); leonardo.dario@humanitas.it (L.D.R.); matteo.colombo@humanitas.it (M.C.); marco.spadaccini@humanitas.it (M.S.); silvia.carrara@humanitas.it (S.C.); marco.giacchetto@humanitas.it (M.G.); vincenzo.craviotto@humanitas.it (V.C.); anita.busacca@humanitas.it (A.B.); 2G.M.E.R.S Medical College & Hospital, Gotri, Vadodara 390021, India; 3Division of Anaesthesiology, Humanitas Research Hospital–IRCCS, Via Manzoni 56, Rozzano, 20089 Milan, Italy; chiara.ferrari@humanitas.it; 4Department of Biomedical Sciences, Humanitas University, Via Rita Levi Montalcini 4, Pieve Emanuele, 20090 Milan, Italy

**Keywords:** endoscopic ultrasound, lumen apposing metal stent, malignant biliary obstruction, LAMS, biliary drainage

## Abstract

This narrative review provides an overview of the application of endoscopic ultrasound-guided biliary drainage (EUS-BD), including EUS-guided gallbladder drainage (EUS-GBD), for the treatment of malignant biliary obstruction. EUS-BD has demonstrated excellent technical and clinical success rates, with lower rates of adverse events when compared with percutaneous trans-hepatic biliary drainage (PTBD). EUS-BD is currently the preferred alternative technique for biliary drainage (BD) in patients with distal malignant biliary obstruction (DMBO) after failed endoscopic retrograde cholangiopancreatography (ERCP). Particularly, this review will focus on EUS-BD performed with the use of lumen apposing metal stent (LAMS). The introduction of these innovative devices, followed by the advent of electrocautery-enhanced LAMS (EC-LAMS), gave the procedure a great technical implementation and a widespread application.

## 1. Introduction

The endoscopic retrograde cholangiopancreatography (ERCP) is widely recognized as the most effective method for achieving biliary drainage (BD) in patients with distal malignant biliary obstruction (DMBO) [1]. However, when malignant conditions affect the distal common bile duct (CBD), it may result in the infiltration and deformation of the ampulla. This can pose challenges or even make it impossible to successfully insert the cannula into the papilla in such instances [2,3,4]. Moreover, there is an increasing number of patients with post-surgical altered anatomies who develop pancreato-biliary diseases requiring BD and for whom ERCP could be technically challenging or even impossible [3,5]. For a considerable period, percutaneous trans-hepatic biliary drainage (PTBD) has been regarded as the established non-surgical alternative for BD in situations where ERCP has been unsuccessful. PTBD is a procedure that is readily available in many healthcare facilities and has proven to be highly efficient. However, it comes with a significant downside, as it is associated with a high morbidity rate and has a notable impact on the quality of life of patients [6,7]. In recent years, there has been a rising trend in the utilization of endoscopic ultrasound-guided biliary drainage (EUS-BD) as a viable alternative method for BD in patients with malignant biliopancreatic disease following unsuccessful ERCP. This approach has demonstrated notable rates of technical and clinical success [8,9]. EUS-BD can be conducted through either hepaticogastrostomy (EUS-HGS) via a trans-gastric intra-hepatic approach or choledochoduodenostomy (EUS-CDS) via a trans-duodenal extrahepatic approach depending on the chosen drainage pathway [10]. Furthermore, EUS-guided gallbladder drainage (EUS-GBD) could be an alternative way for BD to decompress the biliary system, usually used as a rescue approach after ERCP and EUS-CDS failure [11,12]. The idea behind the lumen apposing metal stent (LAMS) emerged from the necessity of designing a device capable of facilitating endoscopic trans-luminal drainage by closely aligning two distinct anatomical structures. LAMS are integrated in a single-step delivery platform and can be deployed under EUS-guidance for the drainage of pancreatic fluid collections, decompression of obstructed biliary ductal systems, establishment of anastomosis or creation of fistulous tracts between organs [13,14,15,16]. With the passage of time, the utilization of LAMS has significantly facilitated the widespread adoption of EUS-BD, especially after the successful implementation of EUS-guided drainage for pancreatic fluid collections. EUS-CDS has been widely regarded by experts as the most efficient and secure method for patients with DMBO following unsuccessful ERCP. In fact, there is a growing recognition of EUS-CDS as a potential first-line therapeutic approach [14,17]. In this review, we will focus on EUS-BD performed with the use of LAMS, including EUS-CDS and EUS-GBD. We will discuss technical aspects, common clinical indications and newer fields of applications.

## 2. Technical Aspects

The procedure is generally performed in an endoscopic room with fluoroscopic equipment and with the patient under conscious sedation or general anesthesia. First, a linear array echoendoscope is advanced until the second duodenal portion. The duodenal bulb is commonly the preferred site to identify the target, namely the CBD. A good diagnostic EUS, including the use of Color Doppler, should always be performed to locate vessels and other structures surrounding the intended puncture site. In the case of EUS-CDS, a dilated CBD (≥15 mm) and a short distance (≤10 mm) between the CBD and the duodenal wall are prerequisites for the correct deployment of the stent. A unique feature of LAMS is their *bi-flanged shape* which gives them the ability to approximate two structures, thereby minimizing the potential risk of leak and dislodgment. Moreover, thanks to their wider lumen, the stents may serve as an access to structures adjacent to the gastrointestinal tract for performing various interventions if required. Over the years, the design of these innovative stents has evolved. Current devices are made of nitinol wire and are fully covered with a silicone membrane which avoids the risk of tissue ingrowth. Nowadays, two types of LAMS are most popular worldwide: the AXIOS stent (Boston Scientific Corporation, Natick, MA, USA) and the NAGI or SPAXUS stents (Taewoong Medical, Gyeonggi-do, South Korea). The deployment of the stent may be performed with the over-the-wire technique or the single-stage technique “freehand”. In the classic technique, the CBD is first punctured with a 19-gauge needle; the injection of contrast confirms the correct target and aspiration of bile can be performed to double check the placement. Then, a 0.025 or 0.035 inch guidewire is inserted into the CBD, and dilation of the fistula, using a cystotome or a dilation balloon, is performed to facilitate the insertion of the stent. The advent of electrocautery-enhanced LAMS (EC-LAMS) allowed a single-step procedure with direct passage of the catheter into the target structure without prior needle access and tract dilation, avoiding device exchange and reducing the procedure time and complexity. The Axios stent was the first electrocautery-enhanced delivery system (EC-LAMS, Hot-Axios, Boston Scientific Corporation, Natick, MA, USA) released into the market, leading to its widespread adoption. Recently, the EC-LAMS version of Spaxus (Hot-Spaxus, Taewoong Medical, Gyeonggi-do, South Korea) has been released into the market and approved for pancreatic fluid collection and GBD [18]. Specifically, for the Hot-Axios system, the EC-LAMS catheter is inserted into the working channel of the echoendoscope and secured to the inlet port of the working channel. The tip of the catheter is positioned tangentially to the CBD. Subsequently, the delivery system is connected to an electrocautery generator and the catheter is introduced into the duct with the application of cautery. When the catheter is fully inside the target structure, the first flange of the stent is deployed, and the confirmation of the correct position is given in real time by ultrasonography. Then, the catheter is slightly withdrawn until the first flange is in contact with the CBD wall. At this point there are two techniques for the release of the second flange. In the first technique, the delivery system is manipulated until a black mark becomes visible at the point where the catheter enters the intestinal mucosa. Then, the system is gently pulled upward to properly position and deploy the proximal flange of the LAMS under endoscopic observation. Alternatively, the intra-channel stent release technique can be utilized, which is considered safer due to its reduced risk of stent dislodgement. In this technique, the stent is inserted into the working channel of the echoendoscope and subsequently pushed out as the echoendoscope is gradually withdrawn (Figure 1) [19,20]. 

## 3. EUS-Guided Choledochoduodenostomy (EUS-CDS)

EUS-CDS entails the establishment of a fistula connecting the CBD and the duodenal bulb, serving the purpose of relieving biliary system compression in instances of distal obstruction. Efficacy and safety of EUS-CDS have been supported by different studies. The first applications of the technique occurred in patients with DMBO after failed ERCP. The first case of EUS-CDS with the use of LAMS was described by Itoi in 2014 [21]. His experience was then followed by other authors in Europe, Asia and the United States, all reporting good rates of technical and clinical success [22,23,24]. Over the years, the EC-LAMS system became more popular, and the single stage technique was demonstrated to have high technical (93.5%) and clinical success rates (97.7%), as well as the over-the-wire technique, with the advantage of time sparing [25]. In a recent extensive multicenter study, conducted across 23 Italian centers, involving 256 patients who underwent EUS-CDS for DMBO after unsuccessful ERCP, remarkably high rates of technical success (93.3%) and clinical success (96.2%) were reported. This comprehensive study encompassed the utilization of both over-the-wire and single-stage techniques. The deployment of the second flange in both techniques was achieved either endoscopically or using the previously mentioned intra-channel stent release technique, as described earlier. The selection of the stent type and size, including Hot-AXIOS and NAGI stents, in EUS-CDS was determined by the endoscopist’s judgment, considering the diameter of the CBD. Several factors were found to be associated with higher rates of technical success. These factors included a larger CBD diameter, the absence of symptoms related to gastric outlet obstruction (GOO), the absence of a pre-existing duodenal stent, and the utilization of the intra-channel stent release technique. The adverse events (AEs) rate was acceptable: 10.5% without any fatal events. 

An interesting finding highlighted by the authors is that technical success, clinical success and the rate of AEs were not dependent on the experience level of the endoscopists. This observation underscores the potential to expand the use of EUS-CDS procedures in real-life settings, even with less experienced endoscopists, as the learning curve for this technique is relatively short [26]. El Chafic et al. suggested that at the end of the procedure, the use of a double pig-tail plastic stent (DPS) or a self-expandable metal stent (SEMS) inserted through the lumen of the deployed LAMS reduce the need for biliary re-interventions maintaining a non-perpendicular LAMS axis within the bile duct, thus avoiding food impaction or sump syndrome (*p* = 0.02) [27]. However, a recent multicenter study on 41 patients, treated with EUS-CDS for DMBO after failed ERCP, showed no significant difference between the strategies of LAMS alone vs. LAMS plus DPS, in terms of recurrent biliary obstruction and AEs rate [28]. Selection of stent type and size for EUS-CDS is mostly based on CBD diameter, at the discretion of the endoscopist. The majority of the data in the EUS-CDS studies present in the literature consider the Hot-Axios system (Boston Scientific, Natick, MA, USA). In particular, small-caliber LAMS (6–8 or 8–8 mm) are usually used for EUS-CDS [29]. However, only three studies report data regarding the direct comparison of the two-stent sizes: 6–8 vs. 8–8 mm stents. In a multicenter study conducted in France, specific attention was directed towards assessing the outcomes of EUS-CDS. The study suggested the utilization of 6–8 mm stents based on the researchers’ extensive experience with this particular stent size. However, it is important to note that the study did not directly compare the effects of different stent sizes on procedural outcomes [30]. No statistically significant difference was observed between 6–8 mm and 8–8 mm stents in terms of technical success, clinical success or stent patency (*p* = 0.661) in a multicenter study including 256 patients [26]. However, in another multicenter study including 120 patients, the overall technical success was similar between patients who had 6–8 mm and 8–8 mm stents, but AEs (OR, 3.71; 95% CI, 1.35–10.19; *p* = 0.008) and reintervention rates (OR, 6.17; 95% CI, 1.22–31.22; *p* = 0.019) were higher in the group of patients with 6–8mm stents (Table 1) [31]. 

## 4. EUS-Guided Gallbladder Drainage (EUS-GBD)

The initial documentation of EUS-GBD can be traced back to 2007, marking the first recorded account of this procedure. This was a case of acute cholecystitis (AC), associated with an unresectable hilar cholangiocarcinoma deemed unfit for surgery, that was treated with the placement of two DPS into the gallbladder with a trans-duodenal approach [32]. Over the years, EUS-GBD underwent technical refinement that minimized the risk of bile leakage and stent migration due to the difficulty in obtaining apposition of the gallbladder and the gastrointestinal wall, thus growing in popularity and widespread application. The first implementation of the technique came from the use of anti-migration tubular SEMS over plastic stents [33,34], followed by the development of LAMS [35] and subsequently EC-LAMS [36]. Currently, the most common indication for EUS-GBD is represented by AC in patients unfit for cholecystectomy [37]. Studies in this population showed that EUS-GBD is a safe and effective technique, with fewer AEs, reinterventions, readmissions and episodes of recurrent cholecystitis, when compared to percutaneous trans-hepatic gallbladder drainage (PT-GBD) [38,39,40,41]. According to these data, ESGE recommends EUS-GBD over PT-GBD for patients with AC at high surgical risk [42]. Furthermore, EUS-GBD can be performed to internalize a previously placed PT-BGD in patients unfit for cholecystectomy, in order to create definitive drainage with encouraging success rates and very limited AEs during the follow-up [43]. The other main indication of EUS-GBD is as a rescue approach for the drainage of patients with DMBO after failed attempts at both ERCP and EUS-BD, which occur in a very small proportion of patients (around 0.1%), as an alternative to performing PTBD [11,42]. While failure of ERCP can be due to various causes, such as altered papillary anatomy, duodenal stenosis or altered post-surgical anatomy [44,45], failure of EUS-BD could be due to reasons such as lack of visualization of the site for biliary access, insufficient dilation of CBD, the interposition of vessels or a distance between the CBD and duodenal wall of more than 10 mm due to tumor infiltration or the presence of a great amount of ascites [46]. EUS-GBD, conceptually similar to a surgical anastomosis [47,48], aims at providing BD through the cystic duct in the case of DMBO. Therefore, cystic duct patency should be assessed accurately before drainage with cross-sectional imaging and a good diagnostic EUS [42,49,50]. To perform an EUS-GBD, a LAMS is placed within the gallbladder to create an anastomosis with either the stomach (cholecystogastrostomy) or the duodenum (cholecystoduodenostomy); moreover, in case of post-surgical anatomical variants the jejunum can be chosen as site of drainage (cholecystojejunostomy) [51]. [Figure 2].

The trans-duodenal approach gives the advantage of a more stable position because it is less affected by the peristalsis; it is also associated with lower risk of food impaction. On the other hand, the trans-gastric approach is preferred in patients who are considered eligible for future cholecystectomy [42]. To limit the risk of food impaction, the use of DPS inserted through the LAMS is recommended. In 2016, Imai et al. published a pioneering study, which involved 12 patients and provided the initial description of EUS-GBD as a rescue approach for patients with obstructive jaundice due to DMBO after ERCP failure. The study reported a technical success rate of 100%, a clinical success rate of 91.7% and an AEs rate of 16.7% [52]. Following this, several studies [11,52] and one meta-analysis [46] evaluated the role of EUS-GBD for DMBO after failed ERCP and EUS-BD. From these studies, pooled rates (95% CI) of clinical success and AEs were 85% (76%, 91%) and 13% (7%, 21%), respectively [46]. Recently, the largest multicenter study on this topic was published, including 48 patients. The authors reported a technical success rate of 100% and a clinical success rate of 81.3%, with a mean total bilirubin reduction after two weeks of 66.5% and an AEs rate of 10.4% [53]. Moreover, EUS-GBD was considered a valid first-line option for BD for low-survival patients with unresectable DMBO, as shown in 37 patients with 100% technical and clinical success rates and an AEs rate of 10.8% [54]. Possible complications of EUS-GBD include bleeding, cholangitis/cholecystitis, bile leak, sepsis, peritonitis, stent migration and perforation. Despite technical innovations enabling EUS-GBD to be performed more efficiently and with greater ease, this procedure continues to be carried out in third-level centers by expert endoscopists. Accordingly, a standardized training program for EUS-GBD and EUS-BD still needs to be developed. Studies suggest that competence is typically attained by endoscopists experienced in interventional EUS after conducting approximately 20 procedures [55,56].

## 5. EUS-BD vs. ERCP in DMBO

As reported by international guidelines and international consensus statements, trans-papillary stent placement via ERCP represents the standard of care for palliation of DMBO [42,57]. However, ERCP is associated with a significant range of post-procedural complications, including pancreatitis, cholangitis, cholecystitis and stent dysfunction, that can in turn lead to the need for reintervention [58,59]. In this context, post-ERCP pancreatitis (PEP) represents the main AE, with serious potential consequences (including death), with a mean incidence of 3.5% but ranging from 1.6% to 15.7% depending on the subset of patients [60]. In consideration of both ERCP-related AEs and the established role of EUS-BD as a means of drainage in case of failed or unfeasible ERCP, new interest emerged in the possible role of EUS-BD as a primary treatment for palliation of DMBO, leading to randomized clinical trials (RCTs) on this subject [61,62,63,64]. In two RCTs, comparable success rates, both technically and clinically, were reported between ERCP-BD and EUS-BD for palliation of DMBO. The studies also found similar safety profiles and rates of reintervention for both procedures [62,63]. Moreover, Paik et al. reported longer duration of patency, lower rates of overall AEs (6.3% vs.19.7% overall; 0% vs. 14.8% for pancreatitis) and reintervention (15.6% vs. 42.6%) and a better preserved quality of life (QoL) for EUS-BD in a study of 125 patients [61]. Finally, EUS-BD does not represent an impediment to subsequent pancreaticoduodenectomy, considering that the site of transmural puncture is also a part of the surgical specimen [23]. Several systematic reviews and meta-analyses were conducted regarding the efficacy of BD by ERCP and EUS-BD for DMBO [65,66,67,68,69,70]. According to those meta-analyses, no significant differences were observed between ERCP and EUS-BD regarding technical and clinical success. Despite that, some meta-analyses highlighted differences in the spectra of complications and the rate of reinterventions. Post-procedure pancreatitis had a significantly higher rate after ERCP than EUS-BD (9.2% vs. 0% [69]); (9.5% vs. 0% [66]); (11.5% vs. 0% [67]). Moreover, EUS-BD was associated with a significantly lower rate of stent dysfunction (mean difference (MD) −0.22%, 95% CI, −0.35–−0.08 [68]; RR 0.12, 95% CI, 0.32–0.91 [67]), tumor in/overgrowth (RR 0.22, 95% CI, 0.07–0.76 [67]; OR 5.35, 95% CI 1.64–17.50 [66]) and reduced risk of reintervention (5.7% vs. 17.5%, OR 0.36, 95% CI, 0.15–0.86 [70]). Overall, from the mentioned meta-analyses it is possible to affirm that EUS-BD has similar success rates as ERCP, while possibly being associated with fewer post-procedural AEs. The promising outcomes observed in studies thus far require further substantiation through additional randomized trials. While awaiting further research, the current evidence from RCTs and meta-analyses has prompted the European Society of Gastrointestinal Endoscopy (ESGE) guidelines to endorse both ERCP and EUS-BD as first-line choices for biliary drainage in situations involving DMBO (Table 2) [42].

## 6. Conclusions

The introduction of LAMS has marked a significant advancement in the field of interventional EUS. Originally designed for fluid collection and drainage, LAMS have swiftly emerged as a fundamental component in numerous EUS-guided procedures, particularly in the context of EUS-BD. Having been firmly established as the preferred method for drainage in patients with DMBO following unsuccessful ERCP, EUS-BD has recently garnered attention as a potential first-line approach for biliary drainage. This recognition stems from its notable achievements in terms of technical and clinical success, as well as its favorable safety profile and manageable learning curve. However, it is imperative to underscore that further rigorous clinical trials are essential to validate the promising evidence currently available on this subject matter.

## Figures and Tables

**Figure 1 diagnostics-13-02788-f001:**
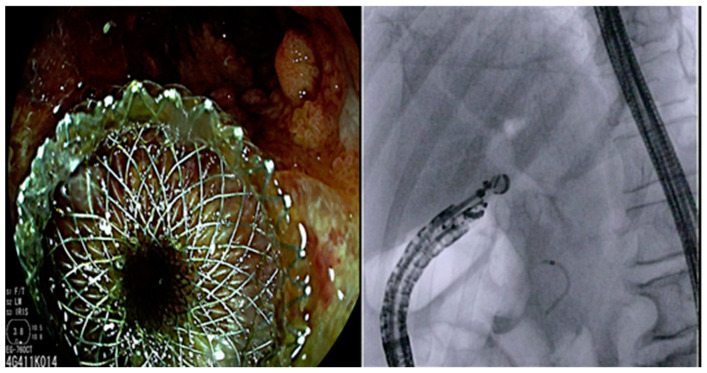
Endoscopic and fluoroscopic image obtained after electrocautery lumen-apposing metal stent (EC-LAMS) deployment in the duodenal lumen.

**Figure 2 diagnostics-13-02788-f002:**
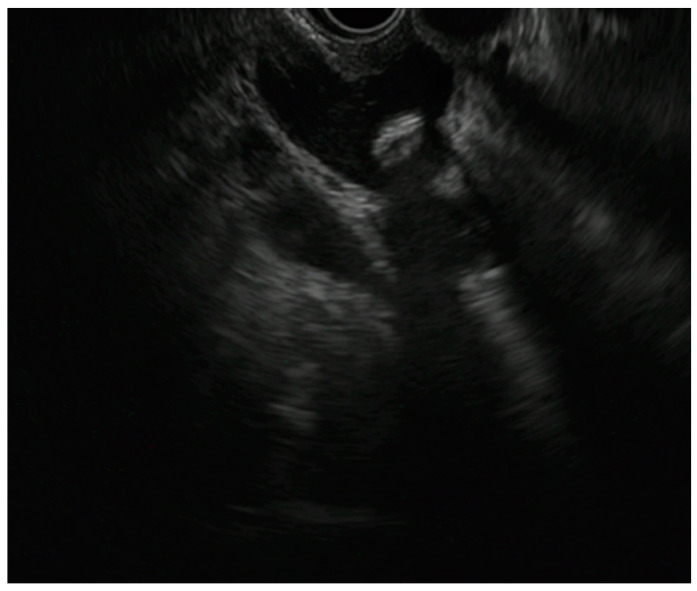
Please revise as Endoscopic ultrasound-guided gallbladder drainage (EUS-GBD) with first flange opening of the elctrocautery lumen-apposing metal stent (EC-LAMS).

**Table 1 diagnostics-13-02788-t001:** Worldwide studies reporting the rates of technical and clinical success of EUS-CDS.

Author [Ref]	Year	Study Type	Patients (n)	TS	CS	AEs	Type of Stent N (%)
Kunda et al. [22]	2016	Retrospective	57	98.2%	94.7%	7%	Axios/Hot-Axios stent 6–8 mm 36 (64.2%) 8–8 mm 2 (3.6%) 10–10 mm 16 (28.6%) 15–10 mm 2 (3.6%)
El Chafic et al. [27]	2019	Retrospective	67	95.5%	100%	6.3%	Hot-Axios stent 10–10 mm
Jacques et al. [24]	2019	Retrospective	52	88.5%	100%	3.8%	Hot-Axios stent 6–8 mm 43 (82.7%) 8–8 mm 7 (13.5%) 15–10 m 2 (3.8%)
Anderloni et al. [25]	2019	Retrospective	46	93%	97.7%	11.6%	Hot-Axios stent 6–8 mm 21 (45.7%) 8–8 mm 19 (41.3%) 10–10 mm 6 (13%)
Jacques et al. [30]	2020	Retrospective	70	97.1%	97.1%	1.6%	Hot-Axios stent 6–8 mm 60 (85.7%) 8–8 mm 9 (13%) 10–10 m 1 (1.3%)
Fugazza et al. [26]	2022	Retrospective	256	93.3%	96.2%	10.5%	Axios/Hot-Axios stent 6–8 mm 86 (33.6%) 8–8 mm 132 (51.6%) 10–10 mm 28 (10.9%) 15–10 mm 7 (2.7%) Nagi stent 12–20 mm 1 (.4%) 12–30 mm 1 (.4%) 16–20 mm 1 (.4%)

**Table 2 diagnostics-13-02788-t002:** RCTs comparing EUS-BD vs. ERCP as first-line approach for palliation of DMBO.

Author [Ref]	Year	Patients (n)	TS%		CS%		AEs%	
			ERCP	EUS-BD	ERCP	EUS-BD	ERCP	EUS-BD
Park et al. [63]	2018	30	100	93	93	100	0	0
Bang et al. [64]	2018	67	94	90.9	91.2	97	14.7	21.2
Paik et al. [62]	2018	125	90.2	93.8	94.5	90	19.7	6.3

## Data Availability

Not applicable.

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
