# Peer review of "Endoscopy Ultrasound-Guided Biliary Drainage Using Lumen Apposing Metal Stent in Malignant Biliary Obstruction"

_diagnostics, 2023, doi:10.3390/diagnostics13172788_

Round 1

Reviewer 1 Report

Please describe whether EUS-CDS for malignant distal biliary obstruction with LAMS is limited to unresectable cases without duodenal obstruction. Also, please state whether EUS-GBD with LAMS will be performed only for unresectable cholecystitis, and whether it will be expanded to treat operable cholecystitis.

Please describe whether EUS-CDS for malignant distal biliary obstruction with LAMS is limited to unresectable cases without duodenal obstruction. Also, please state whether EUS-GBD with LAMS will be performed only for unresectable cholecystitis, and whether it will be expanded to treat operable cholecystitis.

Author Response

thanks for the suggestions. We expanded the discussion accordingly.

Reviewer 2 Report

The article is aimed to review ultrasound-guided biliary drainage using lumen apposing metal stents in malignant biliary obstruction.   The title is “Endoscopy Ultrasound-Guided Biliary Drainage using Lumen Apposing Metal Stents in malignant biliary obstruction”.

1.        This is a review article.

2.        Several factors influence this management.  Please discuss these.

3.        Please review the literature and add more details in the discussion section.

4.        What is the new knowledge of the report?

5.        Please recommend “How to apply this knowledge?” to the readers.

None

Author Response

The article is aimed to review ultrasound-guided biliary drainage using lumen apposing metal stents in malignant biliary obstruction. The title is “Endoscopy Ultrasound-Guided Biliary Drainage using Lumen Apposing Metal Stents in malignant biliary obstruction”.

This is a review article.

Several factors influence this management. Please discuss these.

A: thanks for the suggestion. We better discussed factors affecting patient’s management.

Please review the literature and add more details in the discussion section. 

A: thanks for the suggestion. We expanded the discussion accordingly. We did not report our search strategy, as due for systematic reviews, because of the narrative nature of our review as per invitation. 

What is the new knowledge of the report?  Please recommend “How to apply this knowledge?” to the readers.

Authors: thanks for the suggestions. We better discussed the issue according to your comment.

Here is the revised version of the manuscript with revisions highlighted

Reviewer 3 Report

l   This review provides a comprehensive overview of the advantages, limitations, technical considerations, and potential future directions of EUS-BD using LAMS for malignant biliary obstruction.

l   The comparison between EUS-BD and ERCP for DMBO is addressed, indicating that EUS-BD using LAMS shows similar success rates as ERCP, with potentially fewer postprocedural adverse events. The article concludes by emphasizing the rapid evolution of EUS-BD using LAMS as a valuable intervention, with potential for further exploration and validation through additional clinical trials.

l   In the age of interventional EUS, this review may provide the readers a good overview and detailed understanding of the real role of EUS-BD.

l   For P5 L166: AC n? patients

l   For P5 L172: PTBGD?--> A miss spelling of PTGBD?

Author Response

Thanks for the comments

Dear Referee,

l   This review provides a comprehensive overview of the advantages, limitations, technical considerations, and potential future directions of EUS-BD using LAMS for malignant biliary obstruction.

l   The comparison between EUS-BD and ERCP for DMBO is addressed, indicating that EUS-BD using LAMS shows similar success rates as ERCP, with potentially fewer postprocedural adverse events. The article concludes by emphasizing the rapid evolution of EUS-BD using LAMS as a valuable intervention, with potential for further exploration and validation through additional clinical trials.

l   In the age of interventional EUS, this review may provide the readers a good overview and detailed understanding of the real role of EUS-BD.

Thanks for the support

l   For P5 L166: AC n? We have corrected with AC (acute cholecystitis) in patients

l   For P5 L172: PTBGD?--> A miss spelling of PTGBD? The spelling is percutaneous transhepatic gallbladder drainage (PT-GBD

Round 2

Reviewer 1 Report

No additional comments.